# Utility of Cand PCR in the Diagnosis of Vulvovaginal Candidiasis in Pregnant Women

**DOI:** 10.3390/jof11010005

**Published:** 2024-12-25

**Authors:** Eduardo García-Salazar, Paola Betancourt-Cisneros, Xóchitl Ramírez-Magaña, Hugo Díaz-Huerta, Erick Martínez-Herrera, María Guadalupe Frías-De-León

**Affiliations:** 1Laboratorio de Micología Molecular, Unidad de Investigación Biomédica, Hospital Regional de Alta Especialidad de Ixtapaluca, Instituto Mexicano de Seguro Social para el Bienestar (IMSS-BIENESTAR), Carretera Federal México-Puebla Km 34.5, Ixtapaluca CP 56530, Mexico; eduardogs_01@hotmail.com (E.G.-S.); paola14_02@hotmail.com (P.B.-C.); 2Servicio de Ginecología y Obstetricia, Hospital Regional de Alta Especialidad de Ixtapaluca, Instituto Mexicano de Seguro Social para el Bienestar (IMSS-BIENESTAR), Carretera Federal México-Puebla Km 34.5, Ixtapaluca CP 56530, Mexico; ramaxo@hotmail.com; 3Unidad de Calidad y Riesgo Biológico, Hospital Regional de Alta Especialidad de Ixtapaluca, Instituto Mexicano de Seguro Social para el Bienestar (IMSS-BIENESTAR), Carretera Federal México-Puebla Km 34.5, Ixtapaluca CP 56530, Mexico; hugo180893@gmail.com; 4Programa de Maestría en Ciencias de la Salud, Escuela Superior de Medicina, Instituto Politécnico Nacional, México City CP 07340, Mexico; 5Sección de Estudios de Posgrado e Investigación, Escuela Superior de Medicina, Instituto Politécnico Nacional, Plan de San Luis y Díaz Mirón s/n, Col. Casco de Santo Tomas, Alcaldía Miguel Hidalgo, México City CP 11340, Mexico; erickmartinez_69@hotmail.com

**Keywords:** *Candida* spp., mycotic vulvovaginitis, molecular identification, diagnosis, pregnancy

## Abstract

Vulvovaginal candidiasis (VVC) can lead to multiple complications when it occurs during pregnancy, so it is necessary to diagnose it promptly for effective treatment. Traditional methods for identifying *Candida* spp. are often too time-consuming and have limited specificity and sensitivity. In this work, we evaluated the diagnostic utility of an endpoint PCR assay (Cand PCR) in vaginal swab specimens. Using a cotton swab, 108 vaginal swab samples were taken from pregnant women who consented to participate in the study. The samples were inoculated in Sabouraud agar plates (the gold standard) and subsequently used to extract DNA directly from the exudate. The yeasts isolated from the Sabouraud agar were identified in CHROMagar™ Candida. DNA extracted from vaginal swabs was amplified by Cand PCR. Based on the results of the Cand PCR and the gold standard, sensitivity (S), specificity (E), positive predictive values (PPVs), and negative predictive values (NPVs) were determined. Cand PCR presented an S = 65%, E = 100%, PPV = 100% and NPV = 91%. Cand PCR showed low sensitivity for detecting *Candida* spp. directly from vaginal swabs, but it was useful for identifying the etiologic agent and reducing the time to obtain the result, which is usually at least 48 h.

## 1. Introduction

Vulvovaginal candidiasis (VVC) is an opportunistic infection caused by fungi of the genus *Candida*, which affects the vaginal mucosa with clinical symptoms that include vaginal discharge, edema, itching, pain, irritation, a burning sensation, dyspareunia, and dysuria [1,2]. Worldwide, it represents the second most common vaginal infection among the population of women of reproductive age, reaching an incidence of up to 30% during pregnancy [3]. In addition, it is known that about 75% of women suffer from VVC at least once in their lifetime, and approximately 50% of them will also present a single recurrence [2,4]. In Mexico, according to data from the National Institute of Statistics and Geography (INEGI, https://www.inegi.org.mx/, accessed on 12 November 2024), the population susceptible to VVC (women of reproductive age) represents about 78% of the total female population.

VVC does not pose a risk to the lives of those who present it; however, it is a major public health problem due to the size of the affected population and its negative impact on women’s sexual and emotional lives [5]. In addition, if not properly managed, it can lead to various complications, such as pelvic inflammatory disease, infertility, ectopic pregnancy, pelvic abscess, miscarriage, menstrual disorders, and others, which represent an enormous economic burden for health services [1]. Some studies have shown that the prevalence of *Candida* among pregnant women is higher than in non-pregnant women. Furthermore, it tends to increase with the progression of pregnancy, reaching a peak during the third trimester [6,7]. Some emerging data have also suggested that VVC during pregnancy may be associated with an increased risk of complications, such as the premature rupture of membranes, preterm birth, chorioamnionitis, and congenital cutaneous candidiasis [8]. Therefore, it is recommended that pregnant women receiving prenatal care have rapid and specific laboratory tests for diagnosing VVC [9].

As for the etiology of candidiasis, *C. albicans* continues to be the main causative agent. However, the frequency of non-albicans species has increased, leading to the need to establish a diagnosis based on fungal identification because it impacts the choice of treatment [10]. Despite this, the diagnosis of VVC is usually based mainly on clinical data and yeast isolation and identification through phenotypic methods. Oftentimes, species identification is not feasible due to its cost; therefore, antifungal treatments are usually empirical and unsuccessful in many cases. In addition, it is worth mentioning that many of the phenotypic methods that are widely used in clinical microbiology laboratories have limitations, such as low levels of sensitivity and specificity and high costs; in addition, the results take too long to obtain and depend on the isolation of the pathogen [11]. Due to the limitations of phenotypic testing, molecular technologies based on nucleic acid detection represent more promising tools for the early diagnosis of *Candida* infection [12]. Nonetheless, many molecular assays are not accessible to in-hospital laboratories since the species identification requires a process of enzymatic digestion or sequencing, which makes the methodology more complex [13,14]. Other tests are coupled with automated systems, which represent a considerable investment and require equipment with high operational and maintenance costs [15,16,17,18,19].

Given the lack of diagnostic tools accessible for most in-hospital laboratories in low-income countries, we developed a simplex PCR assay (called Cand PCR) that identifies the eight most frequent *Candida* species in different types of candidiasis with a single pair of oligonucleotides (Cand-F and Cand-R). The species identified by Cand PCR are *C. albicans* 850 bp, *C. glabrata* 1000 bp, *C. tropicalis* 790 bp, *C. parapsilosis* 731 bp, *C. krusei*/*P. kudriazevii* 800 bp, *C. guilliermondii*/*M. guilliermondii* 1100 bp, *C. lusitaniae*/*C. lusitaniae* 590 bp, and *C. dubliniensis* 810 bp. This assay proved adequate in detecting the yeast in blood samples [20]. Based on this prior study, we propose the application of Cand PCR to identify *Candida* spp. directly in vaginal swab samples from pregnant women, without the need to isolate the fungus, to provide a faster and more specific diagnosis that supports therapeutic decisions and, therefore, avoids further complications associated with pregnancy. Hence, the objective of this study was to evaluate the diagnostic utility of Cand PCR in vaginal swab specimens from pregnant women.

## 2. Materials and Methods

### 2.1. Study Design

An observational, descriptive, cross-sectional, and prospective study was conducted in a tertiary hospital in Mexico.

### 2.2. Population

The population consisted of 108 pregnant women who attended routine prenatal visits at the High Specialty Regional Hospital Ixtapaluca (HRAEI) from October to December 2023 and provided vaginal cultures as per medical order. The research protocol was approved by the Research and Research Ethics Committees of the HRAEI (NR-080-2023). All study participants signed the informed consent form.

### 2.3. Selection Criteria

Pregnant women 18 years of age and older presenting vaginal inflammation with discharge, burning, and itching during any trimester of pregnancy were included.

Women receiving antifungal treatment at least 10 days prior to clinical, microbiological, and molecular examination were excluded from the study.

### 2.4. Sample Collection

After informed consent was obtained, a sample of cervical/vaginal mucosa was collected from each patient using a sterile swab with a cotton tip and Stuart transport medium. The samples were taken by the treating gynecologist. The swab with the clinical sample was first used to inoculate the culture medium and then for DNA extraction.

### 2.5. Isolation of Candida spp. from Vaginal Swabs

The swabs with vaginal specimens were used for cultures on Sabouraud dextrose agar (SDA) plates (gold standard). The cultures were incubated at 37 °C for 24–48 h. It should be noted that the gold standard test was performed in the clinical laboratory as part of the patients’ diagnoses.

### 2.6. Phenotypic Identification of Yeasts Isolated from Vaginal Samples

#### 2.6.1. Germ Tube Test

A sample was taken and resuspended from the yeasts grown in SDA in 500 μL of human serum and incubated at 37 °C for two hours. After the incubation period, 20 μL of serum was placed between a slide holder and coverslip for observation under a light microscope (40×) to determine the presence or absence of a germination tube.

#### 2.6.2. Culture on CHROMAgar Candida

On the other hand, yeasts isolated on SDA from vaginal swabs were inoculated in a chromogenic medium (CHROMagar™ Candida, Becton Dickinson, Franklin Lakes, NJ, USA) for yeast identification based on the color of the colonies. This test was performed in the clinical laboratory as part of the patients’ diagnoses.

#### 2.6.3. DNA Extraction from Vaginal Swab Specimens

After inoculating the vaginal swab specimens in SDA, the swabs with the remaining samples were placed in a tube containing 1 mL of sterile PBS-Tween 80 (20%) and were vigorously stirred in a vortex. Afterward, 500 μL was taken and centrifuged at 6093× *g*, removing the supernatant. The sediment was resuspended in 100 μL of lysis buffer, and the instructions of the manufacturer of the Fungi/Yeast Genomic DNA Isolation Kit (Norgen Biotek Corp., Thorold, ON, Canada) were followed. The quality and quantity of the obtained DNA samples were analyzed by spectrophotometry (DS 11 Spectrophotometer, DeNovix Inc., Wilmington, DE, USA) at 260 and 280 nm.

#### 2.6.4. Amplification of DNA Obtained from Vaginal Swab Specimens by Cand PCR

As described by García-Salazar et al. [20], reactions were performed in a final volume of 25 μL, containing the following: 1 ng/µL of DNA extracted from vaginal swabs, 200 μM of dNTPs (Jena Bioscience, Jena, Germany), 1.5 mM of MgCl_2_, 100 pmol of each oligonucleotide (Cand-F and Cand-R) (Sigma-Aldrich, St. Louis, MO, USA), 1 U of *Taq* DNA polymerase (Jena Bioscience), and 1× PCR buffer. As a positive control, 0.5 ng/µL of DNA from the reference strain *C. albicans* ATCC^®^ 18804™ was used, and deionized water (Milli-Q) was used as a negative control. The amplification program consisted of 1 cycle of 3 min at 94° C, 33 cycles of 30 s at 95° C, 30 s at 55° C, 1 min at 72° C, and a final extension of 5 min at 72° C. At the end of the reaction, 6 μL of the electrophoresis amplification products was analyzed in 1.5% agarose gel (Axygen BioScience, Tewksbury, MA, USA) stained with Midori Green (NIPPON Genetics EUROPE, Düren, NRW, Germany) in a 0.5× TBE buffer (45 mM Tris-Base, 45 mM boric acid, 1 mM EDTA, pH 8.3) at 70 V for two hours. The 100 bp DNA Ladder (Promega, Madison, WI, USA) was used as a molecular size marker. The images of the gels were captured by a gel documentation system (Cleaver Scientific Ltd., Rugby, UK). The molecular identification of yeasts was determined based on amplicon size: *C. albicans* (850 bp), *C. glabrata* (1000 bp), *C. tropicalis* (790 bp), *C. parapsilosis* (731 bp), *C. krusei* (800 bp), *C. guilliermondii* (1100 bp), *C. lusitaniae* (590 bp), and C. *dubliniensis* (810 bp).

#### 2.6.5. Evaluation of the Diagnostic Utility of Cand PCR

With the data obtained from the DNA amplification by Cand PCR obtained from the vaginal swab specimens and the results of the gold standard (culture in SDA), the sensitivity (S), specificity (E), and positive (PPV) and negative (NPV) predictive values were determined, as shown in the contingency table (Table 1).

The contingency table consisted of two columns that showed the presence or absence of VVC based on the positive or negative result of the gold standard (culture), respectively. The lines corresponded to the diagnostic test result (PCR Cand). The four cells that were formed corresponded to the true positive (TP), false positive (FP), false negative (FN), and true negative (TN) results, respectively. These terms are defined as follows: true positive—the patient had VVC, and the Cand CRP was positive; false positive—the patient did not have VVC, but the PCR was positive; true negative—the patient did not have VVC, and the PCR was negative; false negative—the patient had VVC, but the PCR was negative. The letters of each cell, as well as the location of the gold standard and the diagnostic test studied, were designated as per convention.

The parameters of sensitivity, specificity, and negative and positive predictive values were calculated using the following formulas:S = TP/[TP + FN] 
E = TN/[FP + TN]
PPV = TP/[TP + FP]
NPV = TN/[FN + TN]

The S, E, PPV, and NPV values were multiplied by 100 to present the results as a percentage.

## 3. Results

During the study period, 142 pregnant women visited the gynecology–obstetrics service, of which 108 met the inclusion criteria and consented to participate in the study. None of the participants requested the revocation of their consent.

### 3.1. Isolation of Candida spp. from Vaginal Swabs

Of the 108 vaginal samples sent for cultures, 23 (21.29%) were positive for yeast. These isolates were labeled by the Lev-number of the vaginal swab sample from which they were isolated; for example, Lev-2 corresponded to the yeast that was isolated from the vaginal swab specimen number 2.

### 3.2. Phenotypic Identification of Yeasts Isolated from Vaginal Samples

#### 3.2.1. Germ Tube Test

Of the 23 yeast isolates obtained from vaginal samples from pregnant women, germination tube development was observed in seventeen isolates (73.9%), while germination tube production was negative in six (26.1%) isolates.

#### 3.2.2. Culture in CHROMAgar Candida

Based on the color of the colonies grown by the 23 yeasts, three species were identified (Table 2), with *C. albicans* (17, 73.9%) being the most frequent species, followed by *C. glabrata* (4, 17.4%) and *C. krusei* (2, 8.7%) (Figure 1).

#### 3.2.3. DNA Extraction from Vaginal Swab Specimens

The quality of the DNA samples extracted directly from the vaginal swab samples (EV-1 to EV-108) was adequate, showing values of A260/A280 in the range of 1.76–2.00. DNA concentrations ranged from 18.7 to 26.2 ng/μL (Figure 2).

#### 3.2.4. DNA Amplification of Vaginal Swab Specimens by Cand PCR

Only 15 (13.8%) of the DNA samples obtained directly from vaginal swabs were positive according to Cand PCR (Table 3). The size of the amplicons observed was 800, 850, and 1000 bp, corresponding to the species *C. albicans*, *C. glabrata,* and *C. krusei*, with *C. albicans* (eleven, 73.3%) being the most frequent species, followed by *C. glabrata* (three, 20%) and *C. krusei* (one, 6.7%) (Figure 3, Figure 4, Figure 5 and Figure 6).

#### 3.2.5. Evaluation of the Diagnostic Utility of Cand PCR

Based on the concordance between the results obtained by the Cand PCR assay and the gold standard, 15 (13.9%) true positives, 8 (7.4%) false negatives, and 85 true negatives (78.7%) were found. There were no false positive results (Table 4). Based on these outcomes, the Cand PCR assay was determined to have a sensitivity of 65.0%, a specificity of 100%, and positive and negative predictive values of 100% and 91.0%, respectively.

## 4. Discussion

During gestation, VVC can lead to various complications that are related to infant mortality and preterm birth [21], which highlights the need for adequate tools for the accurate and timely diagnosis of VVC in pregnant women. The phenotypic methodologies (cultures, biochemical tests, etc.) that are usually used to confirm the diagnosis of VVC have limitations, including the long time required to isolate and subsequently identify the fungus [11]. Taking into account these limitations, different molecular assays have been developed to detect *Candida* and to identify the species directly in vaginal swab samples without the need to isolate the yeast [13,14,15,16,17,18,19,22,23,24,25,26,27]. However, while these assays have adequate sensitivity and specificity, the methodologies they require are often not affordable for many in-hospital laboratories, particularly in low-income areas where infrastructure is minimal and sometimes even obsolete. Therefore, opting for low-cost, methodologically simple, specific, and sensitive molecular assays is more feasible. Such is the case of the Cand PCR assay, which can identify in a single step eight of the most common *Candida* species (*C. albicans*, *C. glabrata*, *C. tropicalis*, *C. parapsilosis*, *C. krusei*, *C. guilliermondii*, *C. lusitaniae*, and *C. dubliniensis*) with a detection limit of 10 pg/μL of DNA or 10^3^ yeasts/mL [20]. In addition, the Cand PCR assay showed sensitivity (73.9%), specificity (96.3%), and positive and negative predictive values (94.4% and 81.2%) suitable for use in the diagnosis of candidiasis using blood samples and bronchoalveolar lavage. However, to use it in other types of biological samples, such as vaginal swabs, it is necessary to evaluate the parameters of S, E, PPV, and NPV since each type of sample has its limitations.

In this study, we analyzed the usefulness of the Cand PCR assay in detecting and identifying *Candida* spp. directly in the DNA obtained from vaginal swab specimens from pregnant women, comparing the results obtained with the gold standard (culture in Sabouraud and identification in CHROMagar™ Candida). The study population included 108 pregnant women with a *Candida* positive rate (gold standard) of 21.3% (23), a frequency similar to that reported in Italy, where the recovery of *Candida* spp. in symptomatic pregnant women was reported to be 25% [28], as well as in Paraguay, where the frequency was 21% [29]. However, the *Candida* positivity rate according to Cand PCR was 13.9% (15), with the following species identified in descending order: *C. albicans*, *C. glabrata,* and *C. krusei*. These results for the identification of *Candida* spp. in vaginal samples were similar to those reported by Trama et al. [23], who, by the amplification and sequencing of the ITS region, found that *C. albicans* and *C. glabrata* were the most prevalent species, followed by *C. tropicalis* and *C. parapsilosis*. Similarly, Fan et al. [27] reported *C. albicans* and *C. glabrata* as the most frequent species, followed by *C. tropicalis* and *C. krusei*. Species identification by Cand PCR was consistent with the results obtained by CHROMagar™ Candida, indicating a specificity of 100% for the assay. However, there was a discrepancy between the Cand PCR and the gold standard regarding the number of samples in which *Candida* was detected. The yeast was isolated in 23 samples according to cultures, while only 15 of the 23 samples were positive according to PCR, which showed the low sensitivity (65%) of the Cand PCR. It should be noted that the sensitivity found is consistent with that reported by Mårdh et al. [22]. They reported 31 samples positive for *Candida* by endpoint PCR from 45 vaginal samples that were positive according to cultures, showing a sensitivity of 68.9%. The relatively low sensitivity is a disadvantage of endpoint PCR assays, as the amplification of minute amounts of DNA may not be visualized with the naked eye in an electrophoretic analysis. In contrast, assays employing other detection methods, such as PCR coupled to quantum dot fluorescence analysis (QDFA PCR), achieve greater sensitivity. This has been demonstrated by Fan et al. [27], who obtained a sensitivity of 88.5% when detecting *Candida* in 340 of 384 samples that were positive according to cultures.

The low sensitivity we found for Cand PCR can also be attributed to other factors. For instance, the quantity of each biological sample used for DNA extraction may have been insufficient since only one swab was collected per patient, and it was prioritized for the cultures, in the same way reported by Payne et al. [16]. Another factor may be related to the presence of inhibitors associated with cotton swabs, as it is known that these can leave fibers or other impurities in a reaction mixture, which could negatively affect nucleic acid amplification [30].

The difficulty in standardizing the detection of yeast from DNA extracted directly from vaginal specimens was also reported by Mårdh et al. [22], who overcame this problem by performing sample dilutions to reduce any potential PCR inhibitors. In the case of Cand PCR, the dilutions did not modify the results.

Although the sensitivity of Cand PCR was not ideal, PPV (100%) and NPV (91%) were adequate. It is essential to mention that both sensitivity and specificity are fundamental operational characteristics for a diagnostic test; however, in clinical practice, what matters is the probability of an individual with a positive result of being indeed sick and the opposite, to know the probability for an individual with a negative result being, in fact, free of the disease [31]. This is known through the positive and negative predictive values, which were greater than 90% for Cand PCR, indicating that the assay may be helpful in diagnosing VVC.

On the other hand, it should be noted that, to date, neither molecular nor conventional methods have been shown to be 100% effective. Therefore, it is crucial to leverage the advantages of each method and combine them to achieve a more specific diagnosis of VVC [32]. This way, Cand PCR could be used simultaneously with conventional methods, such as cultures or microscopy, to diagnose VVC. Lastly, it is worth mentioning that combining methods could help differentiate between colonization and infection, thus avoiding unnecessary treatments [33].

One of the significant advantages offered by Cand PCR over the conventional method (cultures) was that it provided valuable information on the etiology of VVC since it can differentiate quickly (in less than 24 h) and effectively between *C. albicans* and other non-albicans species, such as *C. glabrata* and *C. krusei*, which cause VVC but are clinically indistinguishable from *C. albicans*. The identification of *Candida* at the species level is becoming increasingly relevant in the choice of antifungal treatment, as 50% of *C. glabrata* isolates have a decreased sensitivity to fluconazole, and *C. krusei* is intrinsically resistant to this antifungal [19]. Another advantage is that the endpoint modality in which the Cand PCR is presented makes the test cheaper than real-time PCR tests and, therefore, more affordable to any in-hospital laboratory.

We recognize that the present study has at least three important limitations. The first restraint was that only one swab was collected per patient, which limited the performance of DNA extraction and amplification. The second limitation was that the study only used cotton swabs; perhaps another type of material could have facilitated the collection of DNA directly from vaginal swabs [31]. Finally, another important limitation was that Cand PCR was compared only with the results of CHROMagar™ Candida and not with another molecular method. Therefore, future studies must compare it with other molecular techniques commonly used in clinical practice.

## 5. Conclusions

The Cand PCR assay has a sensitivity of 65%, specificity of 100%, PPV = 100%, and NPV = 91% in the detection and identification of *Candida* spp. from vaginal swab specimens from pregnant women. Although the sensitivity of Cand PCR was not high, the other diagnostic parameters (E, PPV, and NPV) indicate that it can be useful as another resource in the diagnosis of VVC, especially for identifying the etiological agent and reducing the time to obtain the result, which is usually at least 48 h.

## Figures and Tables

**Figure 1 jof-11-00005-f001:**
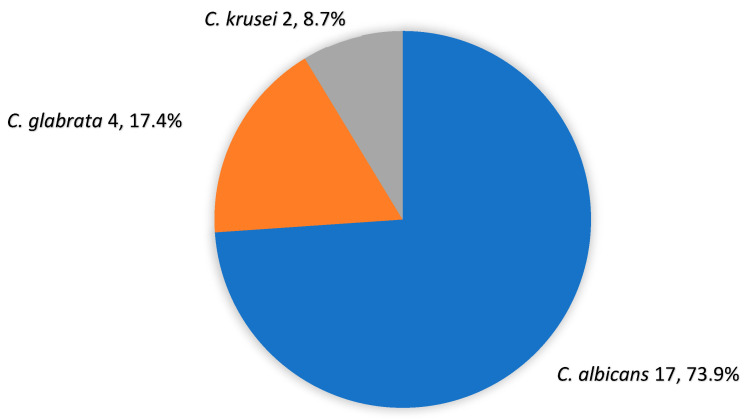
Frequency of *Candida* species isolated from vaginal swabs from pregnant women.

**Figure 2 jof-11-00005-f002:**
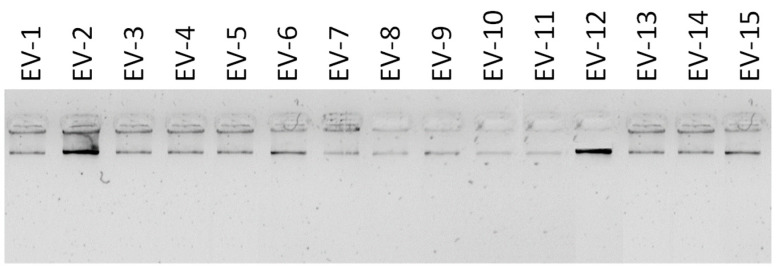
Electrophoretic analysis of DNA samples extracted directly from vaginal swab specimens from pregnant women. Electrophoresis was conducted in 1.0% agarose gel stained with Midori Green in a 0.5× TBE buffer. Electrophoresis was run at 70 V for two hours.

**Figure 3 jof-11-00005-f003:**
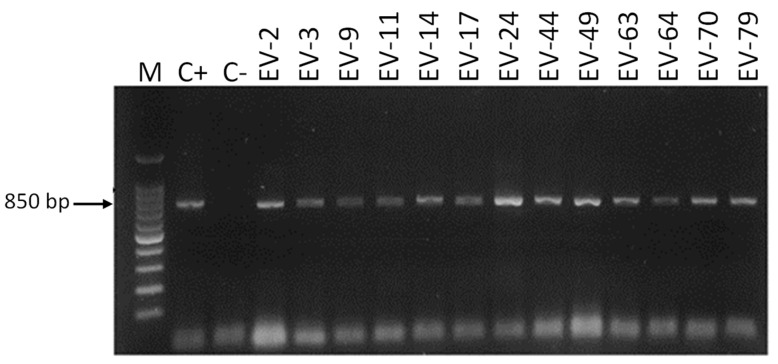
Amplification of DNA samples obtained directly from vaginal swab samples from pregnant women. Electrophoresis was performed in 1.5% agarose gel in a 0.5× TBE buffer stained with Midori Green. M: 100 bp molecular size marker. C−: negative control, C+: positive control (DNA of *C. albicans* ATCC^®^ 18804™).

**Figure 4 jof-11-00005-f004:**
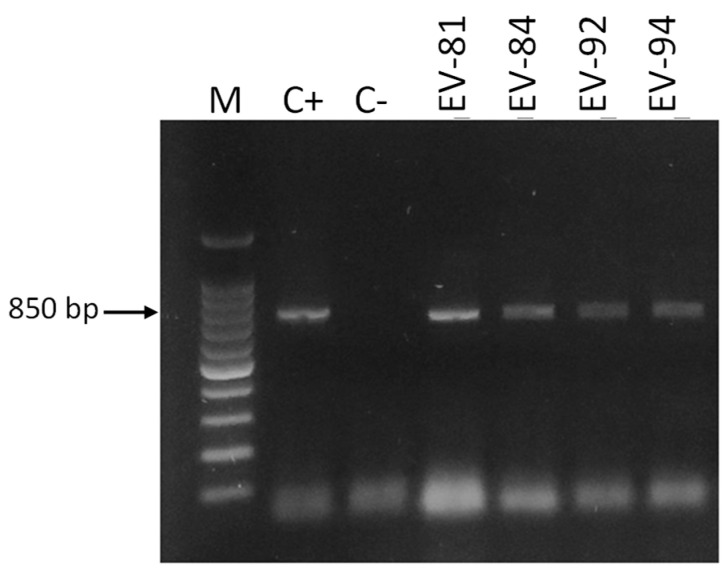
Amplification of DNA samples obtained directly from vaginal swab samples from pregnant women. Electrophoresis was performed in 1.5% agarose gel in a 0.5× TBE buffer stained with Midori Green. M: 100 bp molecular size marker. C−: negative control, C+: positive control (DNA of *C. albicans* ATCC^®^ 18804™).

**Figure 5 jof-11-00005-f005:**
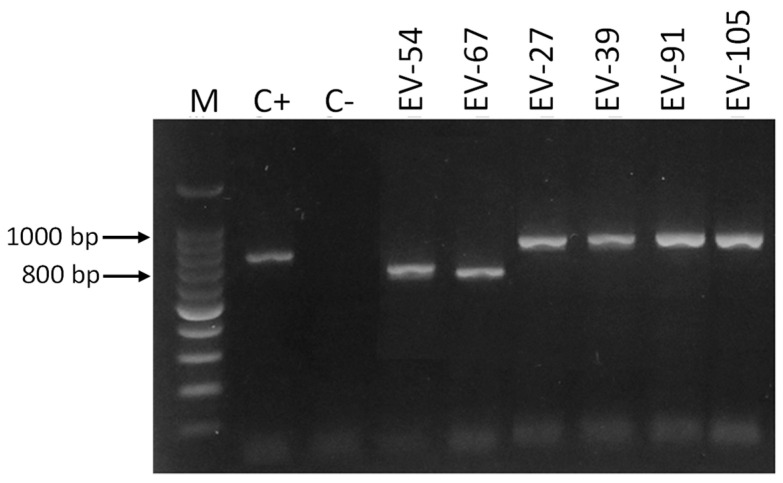
Amplification of DNA samples obtained directly from vaginal swab samples from pregnant women. Electrophoresis was performed in 1.5% agarose gel in a 0.5× TBE buffer stained with Midori Green. M: 100 bp molecular size marker. C−: negative control, C+: positive control (DNA of *C. albicans* ATCC^®^ 18804™).

**Figure 6 jof-11-00005-f006:**
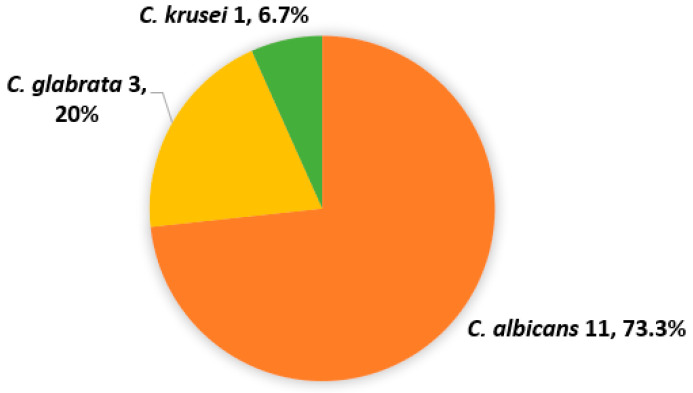
Frequency of *Candida* species identified by Cand PCR in vaginal swab samples from pregnant women.

**Table 1 jof-11-00005-t001:** Association of variables to determine the parameters of sensitivity, specificity, and negative and positive predictive values of the Cand PCR assay in vaginal swab specimens.

Cand PCR Amplification		Gold Standard (Culture)		
		VVC Present	VVC Absent	
	Positive	TP (True Positives)	FP (False Positives)	Total (Subjects with a Positive Test)
	Negative	FN (False Negatives)	TN (True Negatives)	Total (Subjects with Negative Test)
		Total (subjects with the disease)	Total (subjects without the disease)	Total (subjects included)

VVC: vulvovaginal candidiasis; PCR: polymerase chain reaction.

**Table 2 jof-11-00005-t002:** Isolation and phenotypic identification of yeasts isolated from vaginal swab specimens from pregnant women.

Sample No.	Culture			
	Sabouraud	CHROMagar Candida (Color)	Species	Germ Tube Test
2	+	Green	*C. albicans*	+
3	+	Green	*C. albicans*	+
9	+	Green	*C. albicans*	+
11	+	Green	*C. albicans*	+
14	+	Green	*C. albicans*	+
17	+	Green	*C. albicans*	+
24	+	Green	*C. albicans*	+
27	+	Mauve	*C. glabrata*	−
39	+	Mauve	*C. glabrata*	−
44	+	Green	*C. albicans*	+
49	+	Green	*C. albicans*	+
54	+	Pink	*C. krusei*	−
63	+	Green	*C. albicans*	+
64	+	Green	*C. albicans*	+
67	+	Pink	*C. krusei*	−
70	+	Green	*C. albicans*	+
79	+	Green	*C. albicans*	+
81	+	Green	*C. albicans*	+
84	+	Green	*C. albicans*	+
91	+	Mauve	*C. glabrata*	−
92	+	Green	*C. albicans*	+
94	+	Green	*C. albicans*	+
105	+	Mauve	*C. glabrata*	−

**Table 3 jof-11-00005-t003:** Detection and identification of *Candida* spp. by Cand PCR in vaginal swab specimens.

Registration No.	PCR		
	Result	Amplicon Size (bp)	Species
EV-3	+	1000 bp	*C. glabrata*
EV-9	+	850 bp	*C. albicans*
EV-11	+	850 bp	*C. albicans*
EV-17	+	850 bp	*C. albicans*
EV-24	+	850 bp	*C. albicans*
EV-39	+	1000 bp	*C. glabrata*
EV-54	+	850 bp	*C. albicans*
EV-63	+	850 bp	*C. albicans*
EV-67	+	800 bp	*C. krusei*
EV-70	+	850 bp	*C. albicans*
EV-79	+	850 bp	*C. albicans*
EV-84	+	850 bp	*C. albicans*
EV-92	+	850 bp	*C. albicans*
EV-94	+	850 bp	*C. albicans*
EV-105	+	1000 bp	*C. glabrata*

EV: Vaginal swab specimens.

**Table 4 jof-11-00005-t004:** Comparison of Cand PCR results with gold standard (culture) results.

Gold Standard (Culture)				
		VVC Present	VVC Absent	
Cand PCR	Positive test	True Positives: 15	False positives: 0	Total of subjects with a positive test: 15
	Negative test	False negatives: 8	True negatives: 85	Total of subjects with a negative test: 85
		Total of subjects with the disease: 23	Total of subjects without the disease: 85	Total of subjects included: 108

VVC: vulvovaginal candidiasis.

## Data Availability

The original contributions presented in this study are included in the article. Further inquiries can be directed to the corresponding author.

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
