# Peer review of "Utility of Cand PCR in the Diagnosis of Vulvovaginal Candidiasis in Pregnant Women"

_jof, 2024, doi:10.3390/jof11010005_

Round 1

Reviewer 1 Report

The work is devoted to finding new methods for diagnosing candidiasis. Candidiasis is common in pregnant women, which can pose a danger to the fetus. Timely and, most importantly, adequate treatment can reduce the risk of complications. In this regard, the search for reliable and fast diagnostic methods is an urgent task. Dear authors, the work is written in good language, easy to read, the material is presented clearly. In fact, I have only one global question about the work. Of the 8 false-negative samples, 7 contained Candida ablicans. Is it possible that this type of Candida has variability in the region of the genome that is used for PCR detection? And is this the reason for the low sensitivity of the method?

Several questions about the materials and methods.

Line 129 Please indicate the type of microscope that was used. Phase contrast? Light? And if this work was not performed by the authors of the article, this should also be indicated.

Line 140 Please indicate the centrifugation speed in g or provide the type of rotor used.

Lines 148-150 Both concentrations of the substance and their amounts are indicated. Please provide all values ​​consistently.

Line 189-192 The formulas for calculating the values ​​are written incorrectly. Use brackets to show that addition occurs first, then division.

Throughout the text. The use of the abbreviation DNAs is unclear. It is not generally recognized and is not introduced in the text. The term "DNA samples" can be used.

Questions about the figures.

In Figure 1, absolute values ​​must also be indicated.

In Figure 3, the "850 kb" marker is out of place.

Figure 6 will be perceived much better if it is done in the same style as Figure 1.

Author Response

The work is devoted to finding new methods for diagnosing candidiasis. Candidiasis is common in pregnant women, which can pose a danger to the fetus. Timely and, most importantly, adequate treatment can reduce the risk of complications. In this regard, the search for reliable and fast diagnostic methods is an urgent task. Dear authors, the work is written in good language, easy to read, the material is presented clearly. In fact, I have only one global question about the work. Of the 8 false-negative samples, 7 contained Candida ablicans. Is it possible that this type of Candida has variability in the region of the genome that is used for PCR detection? And is this the reason for the low sensitivity of the method?

Answer: This question is very interesting because we cannot deny this possibility (Korabecná, M., Liska, V., & Fajfrlík, K. Primers ITS1, ITS2, and ITS4 detect the intraspecies variability in the internal transcribed spacers and 5.8S rRNA gene region in clinical fungi isolates. Folia Microbiol. 2003; 48(2):233–238. https://doi.org/10.1007/BF02930961). However, it is also important to mention that, during the design of the oligonucleotides, we analyzed several sequences of the same species, and we did not find intraspecies variability. Furthermore, when we amplified the DNA obtained directly from yeasts belonging to the same species, we did not encounter problems, meaning that the specificity was good. In fact, other studies based on the same amplification region have also not found intraspecific variability in Candida albicans, Candida guilliermondii, Candida tropicalis, Candida krusei, Candida glabrata, Candida parapsilosis (Turenne CY, Sanche SE, Hoban DJ, Karlowsky JA, Kabani AM. Rapid identification of fungi by using the ITS2 genetic region and an automated fluorescent capillary electrophoresis system. J Clin Microbiol. 1999 Jun; 37(6):1846-51. Doi: 10.1128/JCM.37.6.1846-1851.1999. Erratum in: J Clin Microbiol 2000 Feb; 38(2):944.) For this reason, we attribute false negative results to low sensitivity since the amount of yeast present in the different samples varies. Thus, it is possible that we will not be able to observe the amplification through an endpoint PCR format.

Several questions about the materials and methods.

Line 129 Please indicate the type of microscope that was used. Phase contrast? Light?

Answer: A light microscope was used; this information was added on line 128.

And if this work was not performed by the authors of the article, this should also be indicated.

Answer: This analysis was conducted by the authors. In the clinical laboratory, only Sabouraud and CHROMAgar Candida cultures were performed.

Line 140 Please indicate the centrifugation speed in g or provide the type of rotor used.

Answer: The centrifugation speed was reported in g.

Lines 148-150 Both concentrations of the substance and their amounts are indicated. Please provide all values ​​consistently.

Answer: All values were standardized and presented as concentration.

Line 189-192 The formulas for calculating the values ​​are written incorrectly. Use brackets to show that addition occurs first, then division.

Answer: The formulas were typed correctly.

Throughout the text. The use of the abbreviation DNAs is unclear. It is not generally recognized and is not introduced in the text. The term "DNA samples" can be used.

Answer: The term "DNAs" was changed to "DNA samples".

Questions about the figures. In Figure 1, absolute values ​​must also be indicated.

Answer: Absolute values were included in Figure 1.

In Figure 3, the "850 kb" marker is out of place.

Answer: "850 bp" was placed in the correct position.

Figure 6 will be perceived much better if it is done in the same style as Figure 1.

Answer: Figure 6 is presented in the same style as Figure 1.

Reviewer 2 Report

See my comments above

Did the investigators carry out any subsequent studies that might shed light on the low sensitivity of the CandPCR ? For instance did they try the DNA extraction process first

See above

Author Response

Is the research design appropriate and are the methods adequately described?

No

It isnt clear from the initial description how the investigators handled the swab. It seems that only one swab was taken per patient and this was used first to inoculate the culture medium and second for DNA extraction. It needs a simple phrasing to make this clearer if this interpretation is correct

See my comments above

Answer: The interpretation is correct. The wording has been changed to improve clarity.

Did the investigators carry out any subsequent studies that might shed light on the low sensitivity of the CandPCR ? For instance did they try the DNA extraction process first

Answer: When Cand PCR was designed, the sensitivity of the assay was analyzed, showing that it could detect up to 10 pg/μL of DNA or 103 yeasts/mL (García-Salazar, E.; Acosta-Altamirano, G.; Betancourt-Cisneros, P.; Reyes-Montes, M.d.R.; Rosas-De-Paz, E.; Duarte-Escalante, E.; Sánchez-Conejo, A.R.; Ocharan Hernández, E.; Frías-De-León, M.G. Detection and Molecular Identification of Eight Candida Species in Clinical Samples by Simplex PCR. Microorganisms 2022, 10, 374. https://doi.org/10.3390/ microorganisms10020374). However, it is not the same thing to work with pure yeasts as a clinical sample. Thus, before embarking on the present work, we conducted some tests to determine if detecting the DNA extracted from swabs contaminated with different yeast dilutions by PCR endpoint was possible, and we encountered no issues. We are currently working on DNA extraction using two swabs per patient, using swabs of other materials (not cotton). Additionally, we are including a DNA carrier (glycogen) to avoid the possible loss of genetic material during the extraction process. We are also testing other types of extraction using Chelex-100. The goal is to provide a simple, low-cost method for diagnosing candidiasis in hospitals or laboratories with limited infrastructure.

Reviewer 3 Report

Review

Title: Utility of Cand PCR in the diagnosis of vulvovaginal candidiasis in pregnant women

The article of Garcia-Salazar describes and analyze a PCR for the detection of Candida species in vaginal swab.

Despite several limitations that authors almost covered in their discussion (mainly sensitivity issues), this study provides new insight on a useful clinical tool, which clinicians might rely on in their clinical practice. I believe that this tool should be furtherly investigate and applied in every clinical context from resource-limited to large healthcare facilities as it could fasten the diagnosis of vulvovaginal candidiasis. However authors are strongly encouraged to address the following points, among all the Gold Standard. From my standpoint, comparing molecular results with a culture-based medium is fairly acceptable, this is the major issue detected.

Comments:  

Introduction.

Line 54-56. Please add a reference at the end of this sentence.

Line 83-86. Please provide the list of Candida species identified by the method.

Method.

Line109. Why were women treated with antimicrobials excluded? An antibiotic treatment could favor the occurrence of vaginal candidiasis. Authors should better elucidate this point giving detailed description of the antimicrobial treatment considered to exclude patients.

Line 125-135. If I understand correctly authors have compared the results of the PCR method with a cultur-based method (CHROMAgar). This represents the major issue with the article. Do the authors have another molecular method able to discriminate yeasts on a species level? VITEK, MALDI TOF, ATR-FTIR, other PCR, sequencing of ITS? How is the result given in normal routine workflow? I would recommend first and foremost to provide a reference molecular method as the Gold Standard if possible and perform the same analyses done with CHROMAgar. If authors are not able to provide such analysis, they must acknowledge this a major study limitation in the discussion section. Authors should also state that future research must compare Cand PCR with other molecular techniques currently used in clinical practice or better with genomic sequencing.

 Results. Results are clear, well presented detailed and correctly elaborated. Figures and tables are clear, well-explained and could be easily interpreted as stand-alone items

Discussion. It is wide, covers most issues and limitations giving explanations and interpretation. Still Sensitivity is extremely low. Even if authors have fully addressed this issue doubts and concerns are raised in the mind of the reader whether this method could actually be a strong support to clinical practice. However, considering that is still a method under development future implementations may increase this flaw, still specificity could be considered a point of strength.  

Author Response

Title: Utility of Cand PCR in the diagnosis of vulvovaginal candidiasis in pregnant women

The article of Garcia-Salazar describes and analyze a PCR for the detection of Candida species in vaginal swab.

Despite several limitations that authors almost covered in their discussion (mainly sensitivity issues), this study provides new insight on a useful clinical tool, which clinicians might rely on in their clinical practice. I believe that this tool should be furtherly investigate and applied in every clinical context from resource-limited to large healthcare facilities as it could fasten the diagnosis of vulvovaginal candidiasis. However authors are strongly encouraged to address the following points, among all the Gold Standard. From my standpoint, comparing molecular results with a culture-based medium is fairly acceptable, this is the major issue detected.

Comments:  

Introduction.

Line 54-56. Please add a reference at the end of this sentence.

Answer: The corresponding reference was placed at the end of the sentence.

Line 83-86. Please provide the list of Candida species identified by the method.

Answer: The Candida species identified by the method were included: C. albicans 850 bp, C. glabrata 1000 bp, C. tropicalis 790 bp, C. parapsilosis 731 bp, C. krusei/P. kudriazevii 800 bp, C. guilliermondii/M. guilliermondii 1100 bp, C. lusitaniae/C. lusitaniae 590 bp, and C. dubliniensis 810 bp.

Method.

Line109. Why were women treated with antimicrobials excluded? An antibiotic treatment could favor the occurrence of vaginal candidiasis. Authors should better elucidate this point giving detailed description of the antimicrobial treatment considered to exclude patients.

Answer: The word was corrected. Only those with antifungal treatment were excluded to avoid problems with the culture.

Line 125-135. If I understand correctly authors have compared the results of the PCR method with a cultur-based method (CHROMAgar). This represents the major issue with the article. Do the authors have another molecular method able to discriminate yeasts on a species level? VITEK, MALDI TOF, ATR-FTIR, other PCR, sequencing of ITS? How is the result given in normal routine workflow? I would recommend first and foremost to provide a reference molecular method as the Gold Standard if possible and perform the same analyses done with CHROMAgar. If authors are not able to provide such analysis, they must acknowledge this a major study limitation in the discussion section. Authors should also state that future research must compare Cand PCR with other molecular techniques currently used in clinical practice or better with genomic sequencing.

Answer: We appreciate the observation and totally agree. This situation has been included as an important limitation of the study.

Routinely, in the institution where the study was conducted, the isolation and identification of the yeasts are performed first in Sabouraud and CHROMagar, respectively. If the yeast’s identity is unclear in CHROMagar, then the identification is verified by VITEK 2 Compact®. None of the yeasts that were isolated and identified during the study were corroborated by VITEK 2 Compact®, so we lack this data. We acknowledge that these proceedings may not be the most appropriate; however, this reflects the reality of many in-hospital laboratories in low-income countries. In some cases, only yeast isolation is performed and reported as Candida spp., leading to inadequate treatment and therapeutic failure. This is precisely why our goal is to create affordable alternatives, such as the PCR Cand.

 Results. Results are clear, well presented detailed and correctly elaborated. Figures and tables are clear, well-explained and could be easily interpreted as stand-alone items

Discussion. It is wide, covers most issues and limitations giving explanations and interpretation. Still Sensitivity is extremely low. Even if authors have fully addressed this issue doubts and concerns are raised in the mind of the reader whether this method could actually be a strong support to clinical practice. However, considering that is still a method under development future implementations may increase this flaw, still specificity could be considered a point of strength.  

Answer: We appreciate this viewpoint. Indeed, it is a specific method that may become relevant, as we have mentioned in the discussion and conclusions, considering that there are differences in the antifungal susceptibility profiles between the different Candida species. Regarding the sensitivity, we continue making adjustments to improve the sensitivity of the PCR Cand.

Round 2

Reviewer 3 Report

Authors have addressed ina a reasonable way all issue and study limitations. One editing must be done in the discussion section "La segunda" must be translated in english

Authors have addressed ina a reasonable way all issue and study limitations. One editing must be done in the discussion section "La segunda" must be translated in english.